# Regenerable Kiwi Peels as an Adsorbent to Remove and Reuse the Emerging Pollutant Propranolol from Water

Jennifer Gubitosa [1], Vito Rizzi [1,*], Paola Fini [2], Sergio Nuzzo [2] and Pinalysa Cosma [1,*]

[1] Dipartimento di Chimica, Università Degli Studi "Aldo Moro" di Bari, Via Orabona, 4, 70126 Bari, Italy; jennifer.gubitosa@uniba.it

[2] Consiglio Nazionale Delle Ricerche CNR-IPCF, UOS Bari, Via Orabona, 4, 70126 Bari, Italy; p.fini@ba.ipcf.cnr.it (P.F.); sergio.nuzzo@ba.ipcf.cnr.it (S.N.)

* Correspondence: vito.rizzi@uniba.it (V.R.); pinalysa.cosma@uniba.it (P.C.); Tel.: +39-0805443443 (V.R. & P.C.)

**Abstract:** This work aims to characterize the adsorption process of propranolol HCl, an emerging pollutant and a widely used β-blocker, onto kiwi peels, an agricultural waste. The use of UV-vis spectroscopy was considered to obtain information about the pollutant removal working in the in-batch mode. In a relatively short time, the adsorption process could remove the pollutant from water. A kiwi peel maximum adsorption capacity of 2 mg/g was obtained. With the perspective of scaling up the process, preliminary in-flux measurements were also performed. The investigation of the whole in-batch adsorption process was conducted by studying the effect of ionic strength (adopting salt concentrations from 0 to 0.4 M), pH values (from 2 to 12), adsorbent/pollutant amounts (from 25 to 100 mg and from 7.5 to 15 mg/L, respectively), and temperature values (from 289 to 305 K). The thermodynamics, the adsorption isotherms, and the kinetics of the adsorption process were also carefully investigated. The Langmuir model fitted the experimental data well, with an $R^2$ of 0.9912, restituting $K_L$: 1 L/mg and $Q_0$: 1.8 mg/g. The temperature increase enhanced the pollutant removal due to the endothermic adsorption characteristics. Accordingly, a $\Delta H°_{298K}$ of +70 KJ/mol was obtained. The pseudo-first-order kinetic model described the process. Due to the results observed during the study of the effects of pH and ionic strength, the prominent presence of electrostatic interactions, working in synergy with hydrophobic forces and H-bonds between the pollutant and kiwi peel surfaces, was successfully demonstrated. In particular, FTIR-ATR measurements confirmed the latter findings. Finally, desorption experiments for recycling 100% of propranolol for each cycle were performed using 0.1 M $MgCl_2$. Ten cycles of adsorption/desorption were obtained and indicated that the percentage of propranolol removal was not affected during each run, increasing the maximum adsorption from 2 to 20 mg/g.

**Keywords:** kiwi peels; emerging pollutants; propranolol; adsorption; green chemistry; recycling

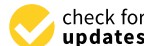



## 1. Introduction

Water is a fundamental resource on Earth, and researchers continuously try to ensure its safety and quality [1]. In particular, dangerous substances, such as textile dyes and heavy metals, have been identified in water for years, and their removal has been attempted [2–9]. It is worth mentioning that, in the last few decades, among the different classes of hazardous chemicals present in water, emerging contaminants (ECs) have been recognized, highlighting another important worldwide problem [10–17]. Pesticides, herbicides, insecticides, fungicides, pharmaceutical compounds, personal care products, etc. are only some examples of ECs [18]. Among these pollutants, particular attention has been devoted to pharmaceutical and personal care products that, although present in water at low concentrations, could induce serious toxicological effects [19]. Furthermore, their persistence and bioaccumulation potential should be considered [19]. Therefore, the need to develop strategies for their removal has risen over the years.

For this purpose, different technologies and methods have been explored, including ion exchange, coprecipitation, filtration, reverse osmosis, Fenton oxidation, biodegradation, ozonation, and electrochemical reduction [20]. However, many disadvantages can be associated with these technologies: (i) the economic aspects, (ii) the complexities of real-life application, and (iii) the use of other chemical reagents, often hazardous, and far from eco-friendly processes [1]. On the contrary, the adsorption process should be preferred, and the advantages derived from its use can overcome some weaknesses. Indeed, it presents, in some cases, weak interactions between contaminants and the adsorbent, lack of selectivity, low surface area, problems related to the adsorbent regeneration and disposal after use, release of unwanted substances ion water, etc. [21]. Despite these disadvantages, it has been extensively used to treat water from ECs, operating as a low-impact strategy [10–17]. Indeed, adsorption is cheap, requires a simple mode of operation, has a simple design, and usually has inexpensive implementation costs from laboratory to industrial scales [22]. The adsorption process has been reported to be useful for removing different classes of pollutants, enlarging its applicability for water treatment [23–26].

Among adsorbents, it is worth mentioning that activated carbon is considered a universal adsorbent and is commonly used to treat contaminated water. However, the high costs associated with its preparation must be considered before real application [27]. On the other hand, a "low-cost" adsorbent material should not require pretreatments, should be abundant in nature, or, as an alternative, could be a by-product or waste material from industries [27]. For example, in the past, several approaches have been studied to develop cheaper and more effective adsorbents containing natural biopolymers, i.e., chitosan and alginate [11,13–15]. Nevertheless, in this context, the use of agricultural wastes, presenting them as a resource with alternative applications, and preventing their disposal, is worth mentioning. In this regard, the work of Rizzi et al. [17] about the removal of tetracycline, an emerging pollutant, through olive pomace, a waste from olive oil production, as an adsorbent is very interesting. Pomace was also proposed in the past for the removal of textile dyes and heavy metals [4,8,9].

Regarding this aspect, the review by Bhatnagar et al. [27] is very interesting and exhaustive as it illustrates, in particular, the use of peels from fruits and vegetables among wastes. Indeed, most fruit peels are discarded as wastes without any application, posing important disposal problems. Not surprisingly, as a renewable resource and an agro-industrial waste, peels have been applied in the treatment of water and wastewaters [28–30]. For example, peels from citrus, orange, pomelo, grapefruit, lemon, banana, cassava, jackfruit, pomegranate, garlic, and others were extensively studied as obtained or as precursors to obtain activated carbon for the removal of both organic and inorganic pollutants [22,27].

Among food/agriculture wastes, the use of kiwi peels as wastes is rather interesting. Their reuse could offer great advantages, obtaining large gains. Not surprisingly, kiwi cultivation occupies a surface area of ~270 × 103 ha around the world, of which ~43 × 103 ha is in Europe. Accordingly, kiwi waste production is high, accounting for ~4.5 × 106 tons globally, with Europe producing ~1 × 106 tons. Consequently, this implies huge global production of kiwi waste with dangerous consequences for the environment, encouraging circular economy approaches as the most pertinent [31].

Interestingly, Sanz et al. [32] stated that different kiwi fruit wastes could be considered resources: peels, pulp, seeds, etc. In addition, kiwi fruit is considered a superfood due to its chemical composition, characteristic flavor, elevated antioxidant and anti-inflammatory properties, and applications ranging from several biomedical fields to nutraceutics [32]. Although these are very attractive benefits, due to the consumption of kiwi fruit, the derived wastes, to our knowledge, have not been valorized, as already reported for other wastes.

By focusing the attention on water treatment, only a few examples [27–30] are accounted for in the literature regarding the use of kiwi fruit. Al-Qahtani [33], in 2016, reported the use of the fruit cortex to remove $Cd^{2+}$, $Cr^{3+}$, and $Zn^{2+}$ ions from an aqueous solution. In 2018, kiwi peels were presented for the removal of nitrate [34]. More recently, in 2019, modified kiwi peels were proposed to remove oil from water [35]. However, in all



of the cited works, the adsorbent was pretreated by using hard work conditions, i.e., by adopting an acid or alkaline medium and, in some cases, high temperatures. Furthermore, besides these works, Rahimnejad et al. [36], in 2018, described the removal of $Pb^{2+}$ by using activated carbon from kiwi peels. More recently, some of the authors of this paper reported using kiwi peels to remove several ECs, with particular attention on ciprofloxacin [31].

However, except for this recent work [31], there are no examples described in the available literature of the use of kiwi peels to remove ECs from water. Therefore, this work aims to further widen the uses of kiwi peels for removing ECs, investigating the mechanism of absorption of another EC, propranolol HCl (PRO). Moreover, strategies to use kiwi peels by lowering the environmental impact and the associated costs, and avoiding any pretreatments [31,33–36] before their use are considered, as already reported in our previous work [31].

PRO was studied because, with other drugs, it is in the top-200 prescribed medications, owing to its wide range of applications [31]. In particular, as a β-blocker, PRO is widely used to treat various cardiovascular diseases. The presence of this drug could arise from hospital effluent, wastewater treatment plants, pharmaceutical industry waste, and excretion after drug administration [37]. Unfortunately, the conventional processes used in traditional wastewater treatment plants are not able to remove PRO efficiently. As a result, PRO can exert harmful effects on aquatic organisms, such as fish, algae, and invertebrates, even at low concentrations. It is highly persistent, and its highest acute and chronic toxicity within the class of β-blockers have been reported [38,39].

Furthermore, it is worth mentioning that a recent study performed by Italian and Australian scientists reported that β-blockers could be potentially used to treat patients affected by COVID-19. In particular, it has been reported that PRO helps to suppress the spread of cancer in the lung, which has an inflammatory profile similar to that of COVID-19. Indeed, patients affected by COVID-19 suffer from many diseases, such as inflammation, because the SARS-CoV-2 virus affects the immune system. In this context, PRO could reduce this inflammation, rebalancing the immune system [39]. Therefore, by also considering the current pandemic situation, the use of PRO will increase over the years, worsening the problem related to water pollution. On this ground, a need to develop effective and, simultaneously, low-cost technologies to remove PRO from water clearly arises. Some studies reported the use of oxidative light-induced processes that, as well-known, suffer from side effects due to the associated costs and the production of secondary pollutants [40,41]. Indeed, the use of advanced oxidation processes to treat water from ECs is well known; however, in the same case, the oxidated by-products are more toxic than the parent molecules [37]. For years, adsorption strategies for PRO removal have been investigated, and Table 1 reports some interesting works available in the literature [37,38,40–45], showing the different classes of employed adsorbents.

**Table 1.** Most interesting and recent articles focused on the removal of PRO from water.

| Adsorbent | Maximum Adsorption Capacity $q_{max}$ mg/g | Reference |
|---|---|---|
| Multi-walled carbon nanotubes | 54.17 | Nie et al. [45] 2020 |
| $Fe_3O_4$/attapulgite magnetic nanoparticles | 12.87–16.87 | Deng et al. [42] 2020 |
| Attapulgite/graphene oxide magnetic ternary composites | 46.8 | Deng et al. [43] 2019 |
| Montmorillonite | 25.9 | Del Mar Orta et al. [44] 2019 |
| Ionic liquid iron nanocomposite adsorbent | 0.1 | Ali et al. [37] 2017 |
| Graphene oxide | 67–77 | Kyzas et al. [38] 2015 |
| Modified attapulgite | 24.56–48.55 | Deng et al. [41] 2011 |
| Kiwi peels | 2 | This work |

In Table 1, the maximum adsorption capacities ($q_{max}$) are reported, denoting that, as a whole, although the values are not very high, they appear more performant than the presented kiwi peels, which show a maximum adsorption capacity of 2 mg/g. However, in favor of kiwi peels' use, it is worth mentioning that the materials proposed in the literature cannot be considered eco-sustainable or classified as low-cost adsorbents. Moreover, only two studies were developed in accordance with the circular economy principles, but used unsustainable approaches. In this regard, Ali et al. [37] reported using ionic liquid iron nanocomposite as an adsorbent to remove PRO, investigating its desorption, and a variety of acids, such as HCl, $HNO_3$, and $H_2SO_4$, for the adsorbent recycling. Although the former can be considered the best one, the working conditions are far from those of the concept of green technologies. Kyzas et al. [38] reported the use of graphene oxide. In this case, the increment in pH and/or the use of organic solvents enabled PRO recovery by proposing, once again, a toxic solvent. However, as reported in our previous works [10–17], the challenge for researchers that work in this field is to find sustainable, low-cost, and efficient processes with the important feasibility of recycling both the adsorbent and the pollutant, according to the principles of Green Chemistry and the Green Economy.

In particular, the Circular Economy has become the focus of a recent major European Union (EU) policy program, and this work reflects the EU Action Plan (European Commission, 2015) that lists biomass and bio-based products as interesting resources (outputs) to be up-cycled to "new input products".

This strategy, referred to as the concept of up-cycling, may offer an interesting alternative to waste dumping, burning, composting, etc. Therefore, a sustainable way to treat waste would lower the environmental impact, with the definite possibility of obtaining high gains in terms of creating novel products that would have positive effects on human health and well-being [46,47].

Therefore, in this context, concerning the past literature [37,38,41–45], the present work proposes PRO and kiwi peels' regeneration by simply using diluted 0.1 M $MgCl_2$, avoiding the use of hard conditions of work or organic solvents, further emphasizing the environmentally friendly use of kiwi peels. Ten adsorption/desorption cycles were studied, preventing the disposal of the adsorbent as secondary hazardous waste material, and recovering 100% of the pollutant, achieving a high gain. A bio-circular economy plan was thus followed, and the demonstrated adsorbent lifetime extension would overcome the problem related to the observed kiwi peels' low adsorption capacity ($q_{max}$ = 2 mg/g) for PRO. Indeed, although the maximum adsorption capacity was low, it could be possible to increase this value to 20 mg/g and above. Moreover, as previously demonstrated in our recent work [31], the proposed adsorbent's main morphological and chemical features were retained after its use and reuse during the adsorption/desorption processes, enabling kiwi peels' use as a durable/recyclable adsorbent material, opening a novel pathway for industrial applications. Indeed, the preliminary in-flux measurements performed during this work confirmed this assessment.

Accordingly, our recent manuscript [31] demonstrated the possibility of removing other contaminants also present in complex mixtures, better encouraging the use of kiwi peels. Furthermore, the material could be considered low cost, and 54€ is supposed to prepare 1 kg of adsorbent. The associated cost is thus very low compared with that of active carbon (around 300€ per kg) and can be easily sustained by industries. The additional positive aspect is attributed to the PRO recovery (the retail cost of PRO is about 30€ per g), which could be reused after appropriate controls.

## 2. Materials and Methods

### 2.1. Chemicals

Kiwi fruits were obtained from local suppliers (Bari, Italy). Propranolol HCl ($C_{16}H_{22}ClNO_2$, M.W. 295.8 g·mol$^{-1}$) was purchased from Sigma-Aldrich (Milan, Italy) and used as received. The propranolol stock solution was prepared by dissolving the powder in deionized water to obtain a concentration of 20 mg·L$^{-1}$ and further diluted according to the case. Concen-

trated HCl and NaOH solutions were used to adjust the pH of the aqueous propranolol solutions when necessary. Salt stock solutions were also used and properly diluted in a propranolol solution, with the aim of assessing the role of ionic strength. Measurements were performed in triplicate, and the standard deviation was reported.

### 2.2. Preparation of the Adsorbent

The fruit peels were placed in hot water until the fruit pulp was almost fully removed. In particular, 20 g of peel was washed with 800 mL of hot water at 353 K and stirred for 15 min. Subsequently, the water was substituted, and the process was continued until clean water was obtained (8 washing cycles were necessary). The obtained peels were dried until constant weight, obtaining the dry adsorbent. Then, the latter was ground to obtain homogeneously shaped and sized peels. Certified test sieves (Giuliani, series ASTM, American Society for Testing and Materials International) were used. Peels with a size of 5 mm × 5 mm were selected and used in this work. Considering only the costs associated with warm water, the supposed cost to prepare 1 kg of adsorbent is ≈50 €.

### 2.3. UV-Visible Analyses

A Varian CARY 5 UV-Vis-NIR spectrophotometer (Varian Inc., now Agilent Technologies Inc., Santa Clara, CA, USA) was used for collecting the UV-vis absorption spectra in a range of 200–500 nm at a 1 nm/s scan rate. The propranolol concentration was determined by monitoring the absorbance intensity at $\lambda$ = 290 nm. For this purpose, a molar absorption coefficient ($\varepsilon$) of 0.017 L·mg$^{-1}$·cm$^{-1}$ was experimentally evaluated by applying the Lambert–Beer law.

### 2.4. ATR-FTIR Spectroscopy Analyses

ATR-FTIR spectra of kiwi peels before and after pollutant adsorption were recorded in a 4000–400 cm$^{-1}$ range using a Fourier transform infrared spectrometer (FTIR Spectrum Two from Perkin Elmer, Waltham, MA, USA), the resolution of which was set at 4 cm$^{-1}$. Sixteen scans were summed for each acquisition.

### 2.5. In-Batch Adsorption Experiments

The kiwi peels' adsorption capacities, $q_t$ (mg·g$^{-1}$), for propranolol were calculated [10,15,17] at different contact times t by using Equation (1) [12–15,17]:

$$q_t = \frac{C_0 - C_t}{W} \cdot V \tag{1}$$

where V is the volume of the propranolol solution (in this case, 15 mL), W is the mass (g) of the dried adsorbent, and $C_0$ and $C_t$ (mg/L) are the amounts of pollutant at time $t_0$ and time t, respectively.

To characterize the role of the propranolol amount in the adsorption process, 25 mg of peels was placed in 15 mL of a propranolol solution at different initial concentrations (7.5 mg/L, 10 mg/L, and 15 mg/L). The process was studied by stirring the pollutant solution in the presence of peels at 250 rpm using a magnetic stirrer. The role of the adsorbent amount was also investigated by using 25, 50, 80, and 100 mg of kiwi peels (fixing the propranolol concentration at 10 mg/L). Therefore, for both cases, the percentage of pollutant removal from water and the peels' adsorption capacities were inferred by collecting the UV-vis absorption spectra at different contact times. In order to obtain information about the nature of the adsorption process, the solution's ionic strength was changed by using different electrolytes at different concentrations—LiCl, NaCl, KCl, MgCl$_2$, and CaCl$_2$ were used for this purpose. Furthermore, the effects of the solution pH (ranging from 2 to 12) and temperature (from 289 to 305 K) on the adsorption process were studied. In this case, the propranolol amount during adsorption was fixed at 10 mg/L, and 25 mg of peels was used.

### 2.6. Kinetics of Adsorption

Kinetic information was obtained by using the pseudo-first-order (PFO) and pseudo-second-order (PSO) kinetic models. For this purpose, Equations (2) and (3) were adopted, respectively [10,15,17].

$$\ln(q_e - q_t) = \ln(q_e) - k_1 \cdot t \tag{2}$$

$$\frac{t}{q_t} = \frac{1}{k_2 q_e^2} + \frac{1}{q_e} \cdot t \tag{3}$$

In both equations, $q_e$ (mg/L) is the adsorption capacity of kiwi peels at equilibrium, $q_t$ (mg/L) represents the adsorption capacity at time t, and $k_1$ (1/min) and $k_2$ (g/(mg·min)) are the rate constants referred to in the adopted models.

### 2.7. Thermodynamic Analysis

The Gibb's Free energy ($\Delta G°$), entropy ($\Delta S°$), and enthalpy ($\Delta H°$) were calculated to study propranolol adsorption on kiwi peels at 289, 298, and 305 K [14,15,17]. In particular, Equation (4) was used to infer the free energy [10]:

$$\Delta G° = -RT \ln K_{eq} \tag{4}$$

where R is the universal gas constant (8.314 J/mol·K), T is the temperature (K), and $K_{eq}$ is the equilibrium constant expressed as $q_e/C_e$. The values of $\Delta H°$ and $\Delta S°$ were calculated by comparing Equations (4) and (5) to infer Equation (6) [10].

$$\Delta G° = \Delta H° - T\Delta S° \tag{5}$$

$$\ln K_{eq} = -\frac{\Delta H^0}{RT} + \frac{\Delta S^0}{R} \tag{6}$$

### 2.8. Isotherms of Adsorption

For our purpose, the isotherms of Temkin, Langmuir, and Freundlich were applied [10]. The Langmuir isotherm supposes that all sites are characterized by constant adsorption energy, so a pollutant monolayer is adsorbed on the adsorbent's surface in the absence of interaction between the pollutant molecules. Equation (7) was applied in its linear form to represent the Langmuir model.

$$\frac{C_e}{q_e} = \frac{1}{K_L Q_0} + \frac{C_e}{Q_0} \tag{7}$$

where $q_e$ (mg/g) is the amount at the equilibrium of adsorbed propranolol, $C_e$ is the related equilibrium concentration expressed in mg/L, $K_L$ is the Langmuir equilibrium constant (L/mg), and $Q_0$ is the adsorbent maximum adsorption capacity (mg/g) value. With respect to the Langmuir model, the Freundlich isotherm describes the adsorbent surface as heterogeneous, with adsorption sites having different energies as a function of the surface coverage. Equation (8) reports the linear form of the Freundlich model.

$$\log(q_e) = \log(K_F) + \frac{1}{n} \log(C_e) \tag{8}$$

where $K_F$ (L/mg) represents the Freundlich constant, and n is the heterogeneity factor. In particular, the 1/n value reveals if the adsorption process is irreversible (1/n = 0), favorable (0 < 1/n < 1), or unfavorable (1/n > 1). Moreover, the Temkin model reports the influence of the adsorbate–adsorbent interactions on the energy of adsorption, suggesting that, during the adsorption process, the heat of adsorption decreases linearly due to interactions between the adsorbent and the adsorbate. A uniform distribution of energies involved

during adsorption is supposed. Equation (9) was used to describe the linearized form of the Temkin model:

$$q_e = B_1 \ln(K_T) + B_1 \ln(C_e) \tag{9}$$

where $K_T$ (L/mol) represents the equilibrium binding constant and $B_1$ refers to the heat of adsorption.

### 2.9. In-Batch Mode Experiments of Desorption

Experiments of adsorbent recycling were performed by using different salt solutions. Among the used salts, $MgCl_2$ was adopted to perform 10 adsorption and desorption cycles. For this purpose, several electrolyte concentrations were explored, and 0.1 M $MgCl_2$ was adopted as the best one. Furthermore, UV-vis absorption spectroscopy was conducted to calculate the amounts of propranolol adsorbed and desorbed. In particular, the adsorbent, after the removal of the pollutant, was washed with water to remove the non-adsorbed PRO, and was then swollen in the salt solution (15 mL) for release.

### 2.10. Zero-Point Charge Determination of Kiwi Peels' Surfaces

The pH of zero-point charge ($pH_{ZPC}$) of the kiwi peels' surfaces was evaluated by adopting a procedure detailed in the literature [10]. In particular, 30 mL of a NaCl solution ($5.0 \times 10^{-2}$ M) was used by changing the pH values from 2 to 12 ($pH_i$) with HCl and NaOH. Therefore, the pH of these solutions was measured before ($pH_i$) and after ($pH_F$) the swelling of 35 mg of kiwi peels (adopted contact time: 48 h, under continuous stirring). The $pH_{ZPC}$ value was obtained by plotting $pH_i$ versus $pH_i$ and $pH_i$ versus $pH_F$, obtaining two lines. The $pH_{ZPC}$ value is the intersection point of these lines.

### 3. Results and Discussion

The UV-vis absorption spectrum of PRO, reported in Figure 1, was used in this work to monitor its removal from contaminated water. For example, Figure 1 indicates that the absorbance bands intensities related to an aqueous solution polluted with PRO 10 mg/L decreased when in contact with 50 mg of kiwi peels. The process was more evident by elapsing the contact time to clearly indicate the adsorption capacity of kiwi peels for PRO sequestering.

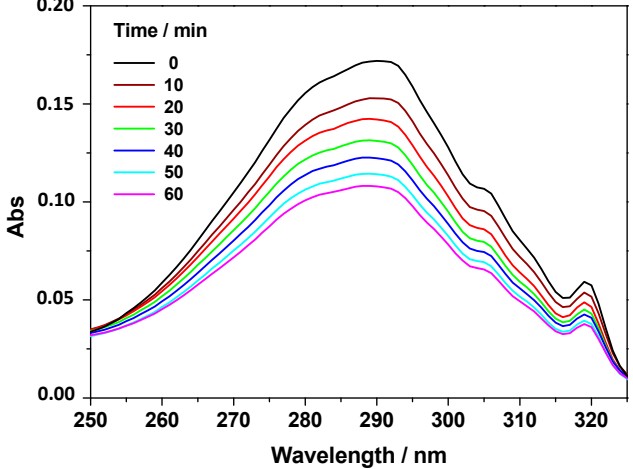

**Figure 1.** UV-vis absorption spectra of 10.0 mg/L PRO solution, pH 5.5, at room temperature (r.t.) in the presence of 50 mg of adsorbent, collected at several contact times.

The adsorption occurred gradually, and, in the first 60 min of contact time, about 50% of the PRO was removed. Furthermore, according to Equation (1), a maximum adsorption capacity $q_{max}$ of 2 mg/g was experimentally evaluated by placing a small amount of kiwi peels in contaminated water to remove PRO until saturation.

Moreover, preliminary in-flux experiments were performed by using a packed column. Indeed, Gubitosa et al. [31] recently demonstrated the possibility of working under dynamic conditions by using kiwi peels. In particular, a commercial syringe-like column (3.0 cm × 5.0 cm) was used for this purpose and filled with kiwi peels, which were previously swollen with water. A solution of PRO (10 mg/L) flowed through the column with the help of a piston. The eluted solution was collected and subjected to UV-vis spectroscopy analysis. By adopting 30 min as the retention time, the percentage of PRO adsorption was calculated, resulting in about 50%, demonstrating the kiwi peels' performance under dynamic work conditions. It is worth mentioning that important counter pressures were absent during the experiments, even if measurements were performed for a prolonged time.

On this ground, aiming to scale up the adsorption process in the future and infer information, this study was mainly devoted to the in-batch mode method, evaluating the effect of several operational parameters on pollutant removal during adsorption.

### 3.1. Roles of PRO and Kiwi Peel Amounts in the Adsorption Process

Different amounts of peels were placed in contact with a PRO solution, maintaining the pollutant concentration constant at 10 mg/L at pH 5.5 and room temperature (r.t.), to infer the role of the active sites available on the kiwi peels' surface in hosting PRO. The efficiencies of adsorption were thus calculated (Figure 2A).

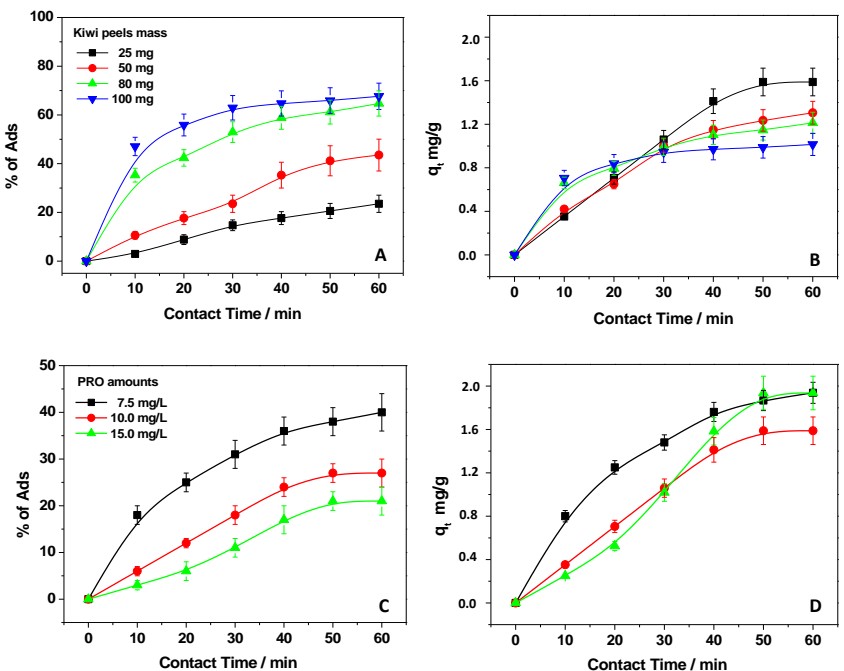

**Figure 2.** Percentage of PRO adsorbed onto kiwi peels, with the corresponding adsorption capacities calculated at different adsorbent weights (**A**,**B**) and PRO concentrations (**C**,**D**).

By increasing the adsorbent amount from 25 mg to 100 mg, the percentage of PRO removal increased, and the effect was more evident at the beginning of the adsorption process, suggesting, as already reported in similar works [10,13,15,17], the key role of free active sites on the adsorbent's surface. In particular, more active sites became available with an increase in the amount of kiwi peels [10,13,15,17]. This observation was confirmed by looking at the $q_t$ values calculated by Equation (1) and reported in Figure 2B. Overall, at the beginning of the process, when all sites were free and available to host PRO, the adsorption capacities increased as the amount of adsorbent increased. Moreover, the plateau region onset, the condition under which equilibrium was achieved, was more quickly reached in correspondence with the greater adsorbent weight. For example, the plateau of the $q_t$

values was reached after 10 min with 100 mg of adsorbent, while it began after 50 min with 25 mg. As already observed by Gubitosa et al. [31], this result indicates that, in the first part of the process, the adsorption sites were not saturated in the presence of a great amount of adsorbent, being available and favoring PRO adsorption. However, interestingly, if, in the first minutes of contact time (0–30 min), the adsorption capacities reflected the trends reported in Figure 2A, evidencing the important role of the available peels' surface, the trends were reversed after 30 min. This observation can be justified considering that, when high amounts of kiwi peels were used, the adsorption capacities joined the plateau rapidly, leveling off the $q_t$ values. In the other cases, since the process was slower, the $q_t$ values achieved the corresponding plateau region more slowly. In excellent agreement with the literature [10,13], this means that, in the presence of great amounts of adsorbent, the adsorption sites were not completely saturated during the process; consequently, the $q_t$ values were lower, although the removal of PRO was higher [13]. The theoretical $q_{max}$ associated with each experimental condition was calculated to understand this concept better. In other words, if the PRO molecules present in the solution were completely adsorbed, the $q_{max}$ values should be 6, 3, 1.87, and 1.5 mg/g with 25, 50, 80, and 100 mg of peels, respectively. The data in Figure 2B show that, only when 80 and 100 mg of adsorbent were used, the obtained $q_t$ values at the plateau region were near the corresponding theoretical $q_{max}$; meanwhile, in the case of 25 and 50 mg, the $q_t$ values at the plateau region were very far from the theoretical ones. These results better explain the observed behavior, confirming that the relative PRO adsorption increased when the PRO/kiwi peel ratio was in favor of the adsorbent, as already observed in Figure 2A. Furthermore, as anticipated before, by looking at Figure 2A,B, both the increment rate of the adsorption percentage and that of the correspondent $q_t$ values increased at the beginning of the process and slowed down with the passing of contact time under all of the considered conditions. This behavior, besides the role of many free sites available at the onset of the process, can also be attributed, as reported in the work of Deng et al. [48] on the removal of emerging pollutants through adsorption, to the presence of repulsive forces between the PRO molecules still in solution and those already adsorbed, which hinder adsorption with prolonged contact times. As observed by Rizzi et al. [10] and Deng et al. [41], as the collision between PRO and free sites was more favored at the beginning of the process, the growth of the $q_t$ values and percentage of adsorption appeared to be reduced [10,48]. As a whole, these results could suggest that, after covering the kiwi peels' surfaces with the molecules of PRO in the first minutes of the contact time, the formation of multilayers could not be significant. In other words, the adsorption was inhibited by elapsing the contact time, indicating the preferable formation of single layers of PRO molecules on the adsorbent's surface and the lack of significant interaction between PRO molecules.

To obtain more information about the nature of the process, the adsorbent weight was fixed at 25 mg, and the amount of PRO was changed, as shown in Figure 2C,D. By decreasing the PRO concentration, the percentage of PRO removal and adsorption capacities increased because more adsorbent surface was available to host the PRO molecules, in agreement with the data discussed so far and other similar works [10,41]. Once again, by considering the hypothetical $q_{max}$ values for each case ($q_{max}$ = 4.5, 6, and 9 mg/g by increasing the PRO initial amount, respectively), the obtained $q_t$ values for more concentrated solutions, referred to as the plateau region, were far from the corresponding relative $q_{max}$. As suggested by Wang et al. [49], the findings indicate that the equilibrium adsorption capacity increased with increasing initial PRO concentration, and the equilibrium time was longer for higher initial PRO amounts [49]. In other words, the increase in the PRO concentration worsened the competition between the PRO molecules and the adsorbent's surface for the limited number of binding sites, reducing the percentage of adsorption and the correspondent $q_t$ values.

### 3.2. Kinetic Analysis

Information about the adsorption process dynamic was obtained through kinetic studies. At first, by following the approaches usually reported in the literature discussing the removal of emerging pollutants by using adsorption techniques, the Weber–Morris intraparticle diffusion model was applied [10,42,43,50,51]. The well-known equation, $q_t = k_{int} \cdot t^{1/2}$, was employed using the $q_t$ values shown in Figure 2. In the equation, $k_{int}$ represents the kinetic constant expressed in mg/(g·min$^{1/2}$), correlated with the intra-particle diffusion rate. Based on the information reported in similar works [10,42–44], the model states that, if the only rate-limiting step is intraparticle diffusion through the adsorbent, the plot of $q_t$ values versus $t^{1/2}$ must restitute a straight line passing through the origin [10,42,43,50,51].

Consequently, the model was applied both when different amounts of PRO (Figure 3A), at constant kiwi peel weight, were in use and when different adsorbent amounts were employed by fixing the PRO concentration (Figure 3B). The results restituted two straight lines that did not pass through zero, as postulated by the model, for both cases. For the sake of comparison and to clarify Figure 3A,B, only two examples of interpolation were reported. As already observed in the literature [10,43,44], intra-particle diffusion cannot be considered the only rate-limiting step. In particular, two or more steps controlled the process [50,51]. In accordance with Lu et al. [50] and Rizzi et al. [10], the adsorption process of PRO on kiwi peels could be divided into two steps. In the first one (Ist Stage), at the beginning of the process, the PRO quickly diffused from the bulk of the solution onto the outer surface of the adsorbent, where the pollutant molecules were subsequently adsorbed. Afterward (IInd Stage), prolonging the contact time, PRO gradually diffused from the outer to the inner surface. During this step, the already adsorbed PRO molecules could hinder further adsorption, slowing down the process, as described previously. As suggested by Lu et al. [50], the presence of a third stage corresponding to the achievement of a dynamic adsorption equilibrium should be expected, but it was not observed for the explored contact times in this study.

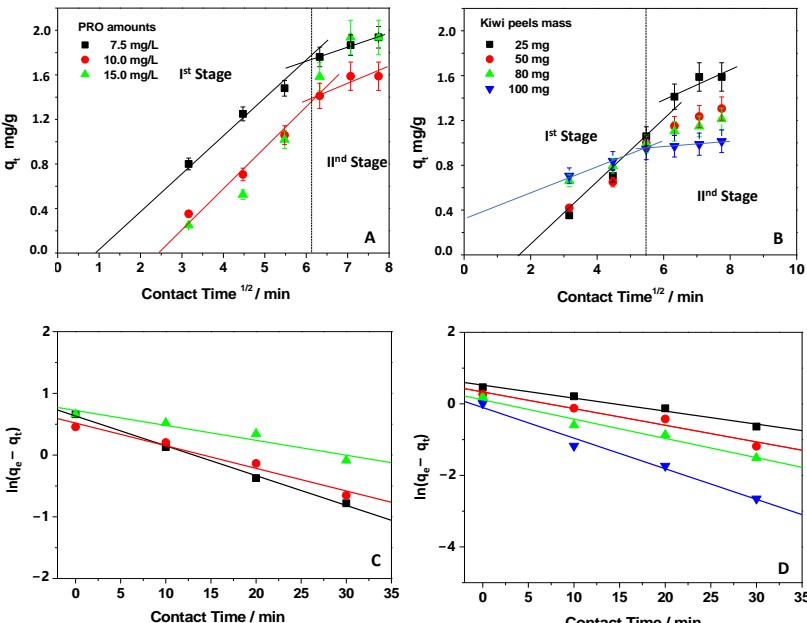

**Figure 3.** Weber–Morris plot of experiments in which the PRO (**A**) and adsorbent (**B**) amounts are changed. Pseudo-first-order kinetic model applied to experimental data in which the amounts of PRO (**C**) and adsorbent (**D**) are changed.

Over, the slope of the Ist stage was greater than that observed for the IInd Stage at every time, indicating, by considering similar behaviors reported in the literature [43,44], that

the former could be considered a minor rate-controlling step [50,51]. However, observing the results reported in Figure 3B, this difference tended to decrease with increasing the kiwi peels' amount. In particular, the slopes of both straight lines, identifying the Ist and IInd stages, slightly differed. This could be interpreted considering that, due to the fast adsorption process under these work conditions, the adsorbent surface was rapidly covered by PRO molecules (Ist stage), and the process appeared slower due to the large contribution of the IInd Stage [45,46].

Furthermore, to gain insight into the kinetic process, and by considering information reported in the literature [47,48], the PFO (Equation (2)) and PSO (Equation (3)) kinetic models were also used, focusing on the first 30 min of contact time.

The analysis was again conducted by changing both the PRO and adsorbent amounts. Therefore, the $q_t$ values were rearranged, and linear fitting was applied to infer which of the kinetic models best described the process. Tables 2 and 3 report the kinetic parameters obtained after linear fitting. The correlation coefficients $R^2$ and the comparison between $q_{e,exp}$ (experimental adsorption capacities at the equilibrium, contact time 60 min) and $q_{e,calc}$ (calculated adsorption capacities, obtained by applying the kinetic equations) were evaluated in accordance with similar works focused on the removal of ECs by means of adsorption [36,37].

**Table 2.** Kinetic parameters for the experiments where the PRO amount changed.

| Concentration (PRO) mg/L | Pseudo-First-Order | | | | Pseudo-Second-Order | | | |
|---|---|---|---|---|---|---|---|---|
| | $q_e^{exp}$ mg/g | $q_e^{calc}$ mg/g | $k_1$ 1/min | $R^2$ | $q_e^{exp}$ mg/g | $q_e^{calc}$ mg/g | $k_2$ g/(mg·min) | $R^2$ |
| 15 | 1.88 | 2.05 | 0.025 | 0.9250 | 1.88 | 2.70 | 0.014 | 0.5532 |
| 10 | 1.63 | 1.68 | 0.040 | 0.9740 | 1.63 | 6.25 | 0.025 | 0.6310 |
| 7.5 | 1.88 | 1.88 | 0.050 | 0.9964 | 1.88 | 4.16 | 0.06 | 0.9960 |

**Table 3.** Kinetic parameters for the experiments in which the adsorbent amount changed.

| Kiwi Peels (mg) | Pseudo-First-Order | | | | Pseudo-Second-Order | | | |
|---|---|---|---|---|---|---|---|---|
| | $q_e^{exp}$ mg/g | $q_e^{calc}$ mg/g | $k_1$ 1/min | $R^2$ | $q_e^{exp}$ mg/g | $q_e^{calc}$ mg/g | $k_2$ g/(mg·min) | $R^2$ |
| 100 | 1.00 | 0.96 | 0.085 | 0.9818 | 1.00 | 1.12 | 0.14 | 0.9994 |
| 80 | 1.23 | 1.18 | 0.055 | 0.9704 | 1.23 | 1.51 | 0.045 | 0.9916 |
| 50 | 1.25 | 1.35 | 0.046 | 0.9557 | 1.25 | 2.43 | 0.010 | 0.9478 |
| 25 | 1.63 | 1.68 | 0.040 | 0.9740 | 1.63 | 6.25 | 0.025 | 0.6310 |

Tables 2 and 3 clearly show that both the $R^2$ values and the comparison $q_{e,exp}/q_{e,calc}$ were better for the PFO than the PSO model. Specifically, when changing the amounts of both the adsorbent and PRO, the correlation coefficients could be considered good when the PFO model was in use; on the other hand, when the PSO model was applied, the $R^2$ values were the worst if the change in PRO was considered, and better for changes in the amount of peels. Interestingly, the comparison between $q_{e,exp}$ and $q_{e,calc}$ failed for the PSO model with respect to the PFO model. Therefore, the PFO kinetic model was considered the best to describe the process, as can be directly observed in Figure 3C,D. This indicates that physisorption mainly controlled PRO removal on kiwi peels through a diffusion process of PRO to the absorbent's surface, which was well accounted for the PFO model [50]. Indeed, the model assumes that the rate-limiting step during adsorption is related to the collisions of the pollutant molecules with available sites at the surface of the adsorbent [52].

### 3.3. Thermodynamic Analysis

The PRO adsorption onto kiwi peels was investigated at different temperature values to obtain the thermodynamic parameters of the process and thus evaluate the temperature

role. For this purpose, 25 mg of adsorbent and a 10 mg/L PRO solution at pH 5.5 were used. Figure 4A,B reports the percentage of PRO removal and the related $q_t$ values, respectively.

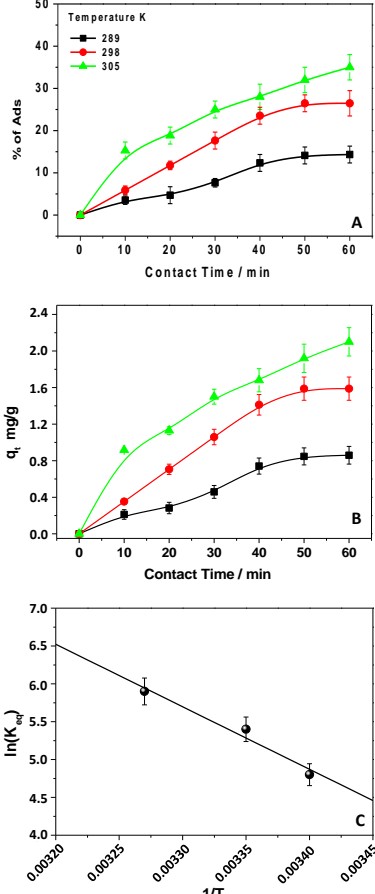

**Figure 4.** Percentage of PRO adsorption (**A**) onto kiwi peels (25 mg) with the related adsorption capacities (**B**) at different temperature values; plot of $\ln(K_{eq})$ vs. $1/T$ (**C**).

The PRO adsorption increased (Figure 4A,B) when increasing the temperature values, thus demonstrating the endothermic characteristics of the process, as already reported in the literature dedicated to describing the removal of emerging contaminants [48]. In particular, temperature's effect was more evident at the beginning of the process, when more free sites were available on the kiwi peels' surfaces to host PRO. Then, the process quickly leveled off. By knowing the $q_t$ values and thus the amount of the non-adsorbed PRO, the $K_{eq}$, corresponding to each examined temperature value, was calculated. Equation (4) was thus applied, obtaining the $\Delta G°$ values reported in Table 4.

**Table 4.** Thermodynamic parameters.

| $\Delta H°_{298K}$ (KJ/mol) | $\Delta S°_{298K}$ (J/mol·K) | $\Delta G°_{289K}$ (KJ/mol) | $\Delta G°_{298K}$ (KJ/mol) | $\Delta G°_{305K}$ (KJ/mol) |
|---|---|---|---|---|
| $+70 \pm 5$ | $+300 \pm 10$ | $-10 \pm 1$ | $-12 \pm 1$ | $-15 \pm 1$ |

The $\Delta G°$ values were negative, suggesting the spontaneity of the process that was more favored with the increase in the temperature values. By applying the linearized form of the Van't Hoff equation (Equation (6)), and thus by reporting $\ln(K_{eq})$ versus $1/T$ in the graph (Figure 4C), the values of $\Delta H°_{298}$ and $\Delta S°_{298}$ were also inferred and are shown in Table 4. $\Delta H°_{298} > 0$ (+70 KJ/mol) better evidenced the endothermic characteristics of the process; on

the other hand, $\Delta S°_{298} > 0$ (+300 J/mol·K) suggested that, as reported elsewhere [10,48], at the adsorbent–adsorbate interface, PRO adsorption induced an increase in the randomness.

### 3.4. Isotherms of Adsorption

The linearized Langmuir, Freundlich, and Temkin models were investigated in this work. For this purpose, Equations (7)–(9) were used, and Figure 5 reports the obtained results related to different PRO concentrations.

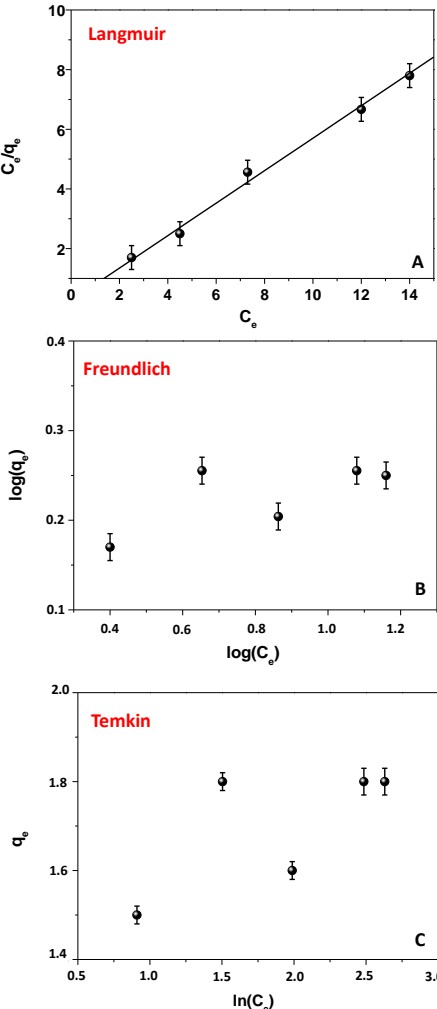

**Figure 5.** Isotherms of adsorption: Langmuir (**A**), Freundlich (**B**), and Temkin (**C**).

As expected, only the Langmuir model fitted the experimental data well, and the result agrees well with all of the observations relative to the behavior of PRO during adsorption. The adherence to the Langmuir model meant that a monolayer of PRO was adsorbed on the kiwi peels' surface without intermolecular interaction between the PRO molecules. Therefore, a finite number of uniform adsorption sites should be considered for PRO adsorption, and the migration of the pollutant through the inner planes of the surface would not be predominant [53]. The same was observed by Deng et al. [41] when modified attapulgite was used. After using the Langmuir model, the following isotherm parameters were obtained: $K_L$: 1 L/mg, $Q_0$: 1.8 mg/g, and $R^2$: 0.9912.

Interestingly, the suitability of the model to fit the data can also be evidenced by considering the $Q_0$ value obtained from the interpolation. $Q_0$ represents, as reported before, the maximum adsorption capacity of the material; in this study, the value restituted by the linear fitting agreed well with the same amount experimentally calculated and reported in Table 1.

### 3.5. Effect of pH

With the aim of investigating the nature of the interaction between PRO and kiwi peels, the role of pH during adsorption was evaluated. PRO solutions, 10 mg/L, with different pH values (from 2 to 12) were studied by adjusting the pH values using concentrated HCl or NaOH. In total, 25 mg of adsorbent was used for each test. As reported in Figure 6A,B, the $q_t$ values of kiwi peels and the corresponding percentage of adsorbed PRO were affected little in the range of pH 4–10, largely decreasing at pH 2 and 12; this is well shown in the inset of Figure 6B reporting the percentage of PRO adsorption calculated at 60 min as a function of pH.

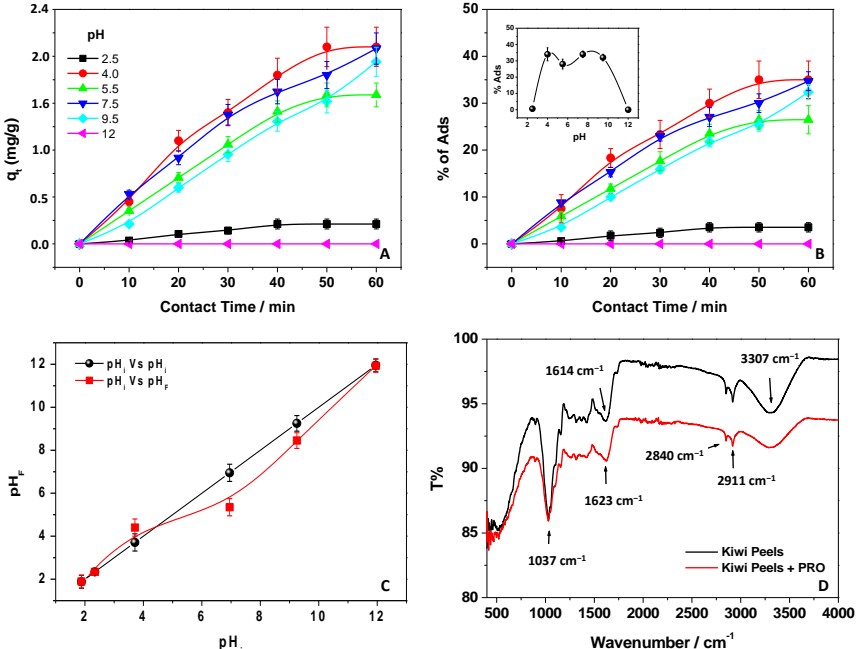

**Figure 6.** Kiwi peel adsorption capacities (**A**) with the related percentage of PRO adsorption (**B**) at several pH values; the inset reports the percentage of PRO adsorption at 60 min; application of the drift method to infer the adsorbent PZC (**C**); FTIR-ATR spectra of kiwi peels before and after PRO adsorption (**D**).

To understand the observed behavior, the PRO molecular features and the PZC of kiwi peels should be considered.

Figure 6C reports that the adsorbent had positive and negative charges at pH < 4 and pH > 4, respectively, suggesting that the $pH_{PZC}$ was at pH 4.

On the other hand, PRO molecules have a $pK_a$ value of 9.5, corresponding to the deprotonation of the PRO amino group ($R_2NH_2^+$) [38,41–44]. Indeed, as reported in other studies [33,36–39], PRO contains a secondary amine group that can be protonated when the environmental pH is lower than its $pK_a$.

PRO can be found as a cation ($PRO^+$) at pH < $pK_a$, and as neutral molecules (PRO) at pH > $pK_a$. On this basis, the observed results can be interpreted considering that, at pH 2, strong electrostatic repulsion between $PRO^+$ and the positively charged kiwi peels occurred, while, at pH 12, with PRO being neutral, adsorption was absent because its hydrophobic characteristics increased [38,41–44]. Therefore, the presence of Coulombian interactions seemed to govern the process. To gain more information about PRO adsorption, another β-blocker was studied and compared. Atenolol was adopted for this purpose.

As reported in Figure 7A, propranolol and atenolol differ in terms of the presence of an extra benzene ring that results in PRO being more hydrophobic than atenolol [38].

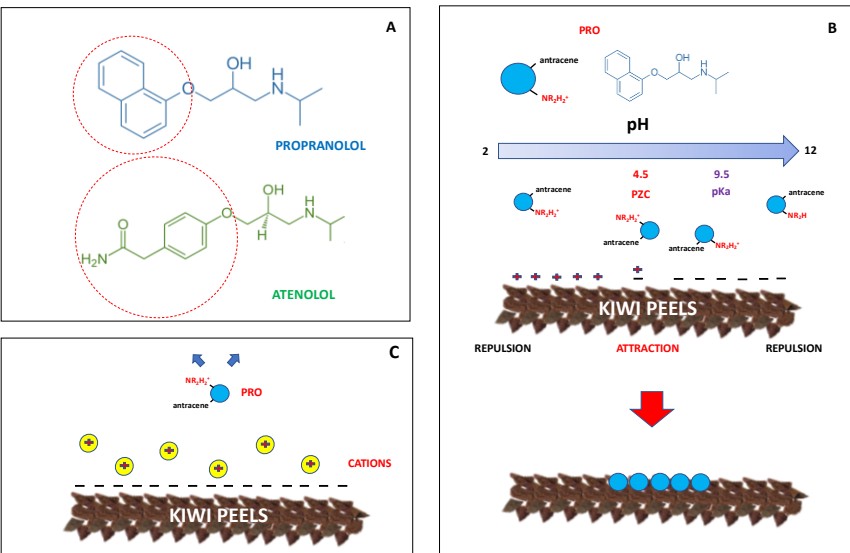

**Figure 7.** Chemical structures of propranolol and atenolol (**A**); interaction between the adsorbent and PRO at several pH values (**B**) and in the presence of cations derived from salts (**C**).

On the other hand, the latter has a second amino group. Very interestingly, atenolol was not removed by using kiwi peels, suggesting that the amino functionality was not the only group involved in the process. Indeed, if the latter consideration is not correct, another amino moiety in the chemical structure of atenolol could enhance the process, which is not the case. Therefore, the anthracene moiety present only in the PRO could play a key role in removing this pollutant from water. The more hydrophobic characteristics of PRO, if compared with Atenolol, favored the interaction with kiwi peels, working in synergy with the electrostatic interaction (a scheme that summarizes the observed behavior is reported in Figure 7B). Therefore, both the anthracene and amino groups were involved in the process, considering thus the presence of Van der Waals' forces, not excluding the contribution of H-bonds between PRO and the kiwi peels' surfaces. In particular, the lack of atenolol adsorption revealed that the Coulombian interaction alone could not be enough to adsorb PRO. Moreover, the absence of PRO adsorption at pH 12, when the pollutant was not charged, suggested that also the hydrophobic interaction alone was not enough.

### 3.6. FTIR-ATR Measurements

FTIR-ATR analyses were performed to gain insight into the nature of the interaction. Since kiwi peel is a lignocellulosic material, it should have specific functional groups, such as alcohols, aldehydes, ketones, carboxylic, phenolic, and ether, that should be taken into consideration during the removal of pollutants. The recent work by Gubitosa et al. [31] discussed these molecular features well. Indeed, the FTIR-ATR signals of kiwi peels shown in Figure 6D indicated how the same IR features occurred on both of the kiwi peels' surfaces (outer and inner surfaces). Briefly, the main IR signals are discussed in the following. If the band at 3307 cm$^{-1}$ revealed the O–H vibration attributed to polysaccharides, phenols, and lignin; on the other hand, the bands at 2840 and 2911 cm$^{-1}$ indicated the C–H bond stretching of the methyl and methylene groups present in the lignin [9,35,36]. The signal at 1614 cm$^{-1}$ can be ascribed to C=O and COO. The weak band observed at 1727 cm$^{-1}$ can be attributed to C=O groups from the carbonyl ester of lignin or hemicellulose ester. The –C–O–C– vibration was observed at 1037 cm$^{-1}$. In regions 1100 and 1500 cm$^{-1}$, weak vibrations attributed to the CH$_3$, –CH$_2$–, C–H moieties, and polyphenolic aromatic ring C=C stretching of lignin were detected. Furthermore, in this region, the bending vibration of O–H bonds from celluloses, the C–O signal in carboxylate groups, and the stretching of esters, ethers, or phenol groups should be considered. Interestingly, after the PRO adsorption, the band at 1614 cm$^{-1}$ shifted to 1621 cm$^{-1}$ and was not well defined. An

interaction of PRO with cellulose OH moieties could be supposed, as already evidenced by Gubitosa et al. [31]. Accordingly, the signal of O–H at 3307 cm$^{-1}$ slightly moved and its relative intensity decreased, indicating the formation of a H-bond with PRO. The aromatic vibrations of lignin were also affected (1100–1500 cm$^{-1}$), with a slight shift of the band at 1037 cm$^{-1}$. All these findings confirmed that, alongside the important presence of electrostatic interactions revealed by the pH effect on the adsorption process, the contribution of H-bonds and hydrophobic forces involving the aromatic rings of lignin and, probably, the anthracene ring of PRO cannot be excluded [9].

### 3.7. Effect of Salts in PRO Solutions

As Deng et al. [41,42] reported in their studies on PRO removal, experiments were performed to better clarify the presence of electrostatic interaction by increasing the ionic strength. Not surprisingly, as is well known [10,36,37], the ionic strength could influence pollutant removal due to competition effects [10,41,42]. Therefore, as the first step, NaCl was adopted at different concentrations. The $q_t$ values and percentage of PRO adsorption were thus inferred (Figure 8A,B). Once again, the experimental conditions were pH 5.5, PRO 10 mg/L, and 25 mg of adsorbent. Figure 8A,B shows that, by adopting a salt concentration that increased from 0.1 to 0.4 M, the PRO removal decreased, and the process was completely blocked in the presence of NaCl 0.4 M. As Deng et al. [41,42] suggested, the competition between the Na$^+$ cations and PRO$^+$ for the same sites explained the decrease in PRO adsorption (an illustrative scheme is reported in Figure 7C).

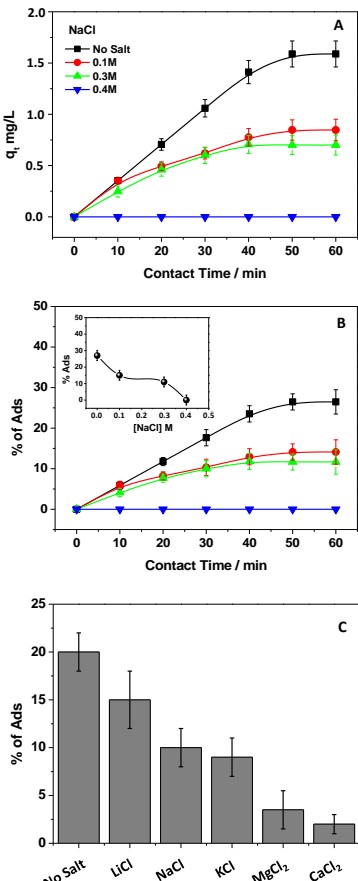

**Figure 8.** Kiwi peels' adsorption capacities (**A**) with the related percentage of PRO adsorption (**B**) calculated at several NaCl concentrations; the inset reports the percentage of PRO adsorption at 60 min; percentage of PRO adsorption onto kiwi peels by changing the cation nature (**C**).

More information was obtained by changing the nature of salts and thus the nature of cations. By selecting 0.1 M as the salt concentration, the electrolyte was changed by fixing

the anion ($Cl^-$) and changing the nature of the cation, i.e., $Li^+$, $Na^+$, $K^+$, $Mg^{2+}$, or $Ca^{2+}$. In this case, the percentage of PRO removal was compared by adopting a contact time of 30 min, and the obtained results are reported in Figure 8C. The PRO removal decreased by increasing the cation size, from $Li^+$ to $K^+$; as already observed by Deng et al. [41,42], by changing the cation charge from monovalent to divalent, when $Ca^{2+}$ and $Mg^{2+}$ were used, the effect was more pronounced.

### 3.8. Desorption of PRO and Adsorbent Recycling

Desorption experiments were performed to assess the recycling of the kiwi peels and PRO. The aim was to operate following circular economy and green chemistry strategies. Since the use of electrolytes reduced PRO removal, inhibiting its adsorption, electrolyte solutions were used to desorb PRO after adsorption. For this purpose, NaCl, $MgCl_2$, and $CaCl_2$ solutions were used by adopting two concentrations, 0.1 and 0.5 M, thus investigating two experimental conditions in which the adsorption was slightly and strongly influenced, respectively, according to Figure 8A. After PRO adsorption on 25 mg of adsorbent with a contact time of 60 min and pH 5.5, the kiwi peels were swollen, under stirring, in the aqueous salt solutions. The percentage of PRO desorption was calculated and is reported in Figure 9A.

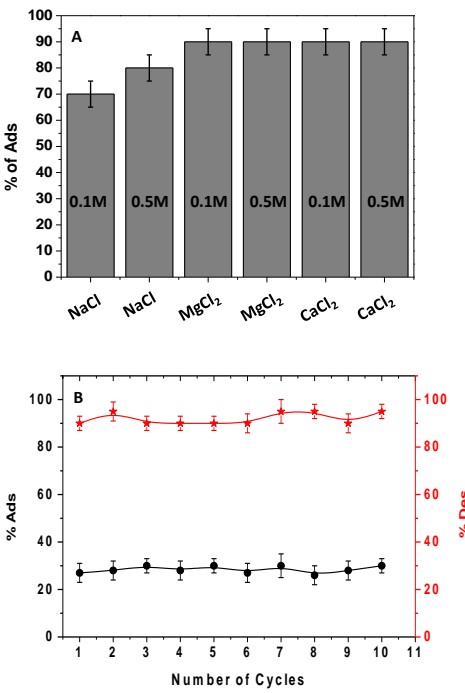

**Figure 9.** Percentage of PRO desorption from kiwi peels in the presence of different concentrations of NaCl, $MgCl_2$, and $CaCl_2$ (**A**); consecutive cycles of PRO adsorption and desorption in 0.1 M $MgCl_2$ (**B**).

When NaCl was used, the results suggested that, by increasing the NaCl concentration from 0.1 to 0.5 M, the percentage of desorbed PRO increased, but it was not complete. The release was more pronounced and almost complete if $MgCl_2$ and $CaCl_2$ were used. Therefore, the PRO release was enhanced, and no dose–response results were obtained under these conditions. According to our previous study [31], 0.1 M $MgCl_2$ solution was selected for the PRO recovery, and several adsorption/desorption cycles were studied. The percentage of the adsorbed/desorbed PRO (contact time: 60 min) is reported in Figure 9B. After 10 adsorption/desorption cycles, the adsorption/desorption process remained very performant, and the same recovery of about 100% of the adsorbed PRO ensured recycling.

## 4. Conclusions

This work reports the use of an agricultural waste, kiwi peels, for propranolol removal from water. The low-cost adsorption process was studied by changing various chemical–physical parameters. By increasing the temperature values, an increase in PRO removal from the water was observed. The process was spontaneous ($\Delta G° < 0$) and endothermic ($\Delta H° > 0$), occurring with an increase in entropy. Isotherms and kinetic models were applied, indicating that the adsorption process occurred with a molecular monolayer formation due to the applicability of the Langmuir isotherm model. The pseudo-first-order kinetic model described the adsorption process well, indicating the kinetic relevance of diffusion in PRO adsorption onto kiwi peels. Furthermore, it was observed that, by increasing the adsorbent amount and diluting the PRO solution, the PRO removal efficiency increased, highlighting the important role of free active sites on the peels' surfaces during adsorption. The influence of the pH values of solutions containing PRO played a key role in the process, suggesting the presence of electrostatic interaction between the pollutant and the adsorbent. Particularly, PRO adsorption occurred with the same efficiency in the pH range of 4–10, and it decreased both at pH 2 and 12, in agreement with the $pK_a$ value of the drug ($pK_a$: 9.5) and the PZC (pH 4.5) of the adsorbent. Repulsion between the peels' surfaces and PRO was observed at extreme pH values. Therefore, interactions between the protonated amino moiety of PRO and the positively charged surface of the peels were suggested. Moreover, atenolol was used, which differs from PRO due to the absence of the anthracene moiety, and a lack of adsorption was observed, indicating the additional presence of H-bonds and hydrophobic forces working in synergy with the Coulombian interaction. FTIR-ATR measurements confirmed the results. Accordingly, the adsorption performed in the presence of inorganic salts corroborated the latter finding, slowing down the process. As a result, it was possible to achieve PRO desorption by using 0.1 M $MgCl_2$ solution, suggesting an ionic exchange mechanism, favoring the recycling of both PRO and kiwi peels for 10 cycles, thereby increasing the adsorbent's lifetime, according to green chemistry and circular economy principles. In conclusion, the use of kiwi peels will not only reduce the environmental impact of waste disposal in favor of green strategies to decontaminate water, but it will also favor a new virtuous alliance between agriculture, enterprise, and research. Therefore, academic research will close the bioeconomic circle, giving rise to promising collaborations that will multiply the value of kiwi fruits while improving the engineering of production processes with the development of new plant technologies for the transformation of kiwi wastes into value-added materials.

**Author Contributions:** Conceptualization, V.R., P.C. and J.G.; methodology, V.R. and J.G.; validation, V.R., J.G. and P.C.; formal analysis, V.R.; investigation, V.R., J.G. and P.F.; resources, P.C. and P.F.; data curation, V.R.; writing—original draft preparation, V.R. and J.G.; writing—review and editing, V.R., J.G. and P.C.; visualization, V.R. and S.N.; supervision, P.C.; project administration, P.C. and V.R.; funding acquisition, P.C. and P.F. All authors have read and agreed to the published version of the manuscript.

**Funding:** This work was supported by the follwing projects: "Research for Innovation (REFIN) per l'individuazione dei progetti di ricerca"—PUGLIA FESR-FSE 2014/2020, Project title: "Incontro tra Ricerca & Impresa per lo Sviluppo Sostenibile del territorio (IRISS): valorizzazione di scarti alimentari per la rimozione di contaminati emergenti dalle acque"; "Dottorati di ricerca in Puglia XXXIII, XXXIV, XXXV ciclo, POR PUGLIA FESR-FSE 2014/2020"; Horizon Europe Seeds, Project title: "Gestione sostenibile di scarti Agroalimentari come fonte Innovativa di biomateriali multifunzionali per la salute umana e l'Ambiente (G.A.I.A.)"; and the LIFE CLEAN UP project, grant number LIFE16ENV/ES/000169.

**Institutional Review Board Statement:** Not applicable.

**Informed Consent Statement:** Not applicable.

**Data Availability Statement:** Not applicable.

**Conflicts of Interest:** The authors declare no conflict of interest.

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
