# Peer review of "Regenerable Kiwi Peels as an Adsorbent to Remove and Reuse the Emerging Pollutant Propranolol from Water"

_processes, doi:10.3390/pr10071417_

Round 1

Reviewer 1 Report

1.       In the introduction, kindly avid the excessive use of lumped citations

2.       The adsorption capacity of 2 mg/g for the pollutant quite low? What justification is there for this? How will this option compete favourably with other adsorbent materials?

3.       Section 2.2, were the peels not grounded and sieved???

4.       Line 178, that number of references are not needed for equation 1 alone. Use only one

5.       Line 207, excessive lumped citation

6.       When discussing effects of parameters, present scientific justifications for observations made. Make comparisons with observations from other studies using other materials

7.       Do a grammar check of the paper

Author Response

Reviewer 1

Comments and Suggestions for Authors

  1. In the introduction, kindly avoid the excessive use of lumped citations

Reply:

The whole manuscript has been carefully revised, and the use of lumped citations has been avoided when possible.

  1. The adsorption capacity of 2 mg/g for the pollutant quite low? What justification is there for this? How will this option compete favorably with other adsorbent materials?

Reply:

We agree with the Referee’s comment. However, the very important aspect of this paper is related to the adsorbent and pollutant recycling, valorizing food wastes according to Green Chemistry and Sustainability principles. The use of Kiwi Peels could offer great advantages by reusing waste and obtaining high gains. Kiwi cultivation occupied a surface of ~ 270 × 103 ha around the world, and Europe contributed with ~ 43 × 103 ha. Accordingly, the Kiwi waste production is high, accounting for ~ 4.5 × 106 tons globally, with Europe producing ~1 × 106 tons. Consequently, this implies a huge global production of Kiwi wastes with dangerous consequences for the environment, encouraging the proposed Circular Economy approach as the most pertinent.

Using a Green and Sustainable approach by exploiting salt solutions has been reported for the adsorbent and pollutant recovery, obtaining another important advantage and further highlighting as the material could compete with other adsorbents. So, although the maximum adsorption capacity is relatively low, it is possible to increase this value by performing several cycles of adsorption and desorption. In this case, we have demonstrated the possibility of performing 10 cycles of adsorption/desorption increasing the maximum adsorption capacity from 2 to 20 mg/g, a value that could be further improved. Indeed, as demonstrated in our previous work (see reference 31), the Peels features appeared unchanged after the recycling, favoring their potential application for industrial purposes (work is in progress about studying these aspects in our laboratory). Moreover, in our recent manuscript (see reference 31), it has been demonstrated the possibility of also removing other different pollutants present in complex mixtures. Furthermore, we are working on the possibility to also adsorb textile dyes and heavy metals for wide-ranging applications.

The associated costs have been evaluated to further evidence the competitiveness of Kiwi Peels with other adsorbents. For this purpose, the following considerations have been taken in mind: (i) the cost associated with the raw adsorbent could be considered null because it is a waste received from agri-food industries; (ii) only the costs associated with the hot washing water, needed for the pre-treatments of Kiwi Peels, should be taken in consideration. To wash 1Kg of Peels, around 300L of water are necessary, and the supposed cost is around 0.42 € (the cost for 1m3 of tap water can supposed be 1.4 €). Furthermore, the important contribution of heat necessary to warm water is further considered (around 0.18€ are needed to heat 1L of water), and 54 € can be necessary to prepare 1Kg of adsorbent. On this bases, the total cost arises, and it can be mainly attributed to the heat necessary for heating water useful for pre-treating the adsorbent before its use. However, the cost is very low if compared, for example, to the cost of active carbon (around 300 € per Kg), and easily sustainable by industries, also considering the additional positive aspect that the adsorbent can be recycled with simple salt solutions. Moreover, another gain can be attributed to the Propranolol (PRO) recovery (the retail cost of PRO is about 30 € per g) that, after appropriate controls, could be reintroduced into the market.

Finally, the use of Kiwi Peels will favor a new virtuous alliance between agriculture, enterprise, and research. So, the academic research will close the bioeconomic circle, giving rise to promising collaborations that will multiply the value of Kiwi fruits while improving the engineering of production processes with the development of new plant technologies for the transformation of Kiwi wastes into value-added materials. In turn, this innovation will contribute to the enhancement of the competitiveness of Italian producers and to revitalize the territory.

In our opinion, these are all positive aspects that can overcome the low adsorption capacity of the material, together with the fact that in the past literature, the information about the use of Kiwi Peels for the removal of emerging pollutants is completely missing, once again evidencing the importance of this work.

These considerations have been better emphasized in the revised manuscript.

  1. Section 2.2, were the peels not grounded and sieved???

Reply:

Section 2.2 has been revised accordingly. In detail, Kiwi Peels were placed in hot water until the fruit pulp was removed and left to dry in an oven at 50°C. Subsequently, the latter were grounded to obtain homogeneous shaped and sized Peels. Certified test sieves (Giuliani, series ASTM, American Society for Testing and Materials International) were used for this purpose. Peels having a size of 5 mm x 5 mm were selected and used during this work.

  1. Line 178, that number of references are not needed for equation 1 alone. Use only one

Reply:

The paper has been revised accordingly.

  1. Line 207, excessive lumped citation

Reply:

We apologize for this inconvenience. It is an error due to the use of software (Mendeley) to insert references. The manuscript has been revised accordingly.

  1. When discussing effects of parameters, present scientific justifications for observations made. Make comparisons with observations from other studies using other materials

Reply:

Thank you for this suggestion useful to improve the quality of the paper. The manuscript has been revised accordingly.

  1. Do a grammar check of the paper

Reply:

The whole manuscript has been revised accordingly.

Reviewer 2 Report

Abstract: Propranol • HCl –This is an error?

Abstract lacks of numerical values that are important.

How PRO is treated now and what is the limitations of the treatment mode? Please elaborate in Introduction.

Section 2.2 is simple. Please elaborate how the adsorbent was prepared in detail. Grinding, size, etc.

Eq 2 and 3 the K should be k.

Please check other Eqs carefully and please refer these papers and cite if needed (you can refer other papers too): Artificial neural network (ANN) for modelling adsorption of lead (Pb (II)) from aqueous solution.T Khan, MRU Mustafa, MH Isa, TSBA Manan, YC Ho, JW Lim, NZ Yusof Water, Air, & Soil Pollution 228 (11), 1-15

Performance of branched polyethyleneimine grafted porous rice husk silica in treating nitrate-rich wastewater via adsorption. ND Suzaimi, PS Goh, NANN Malek, JW Lim, AF Ismail. Journal of Environmental Chemical Engineering 7 (4), 103235

Enhancing the performance of porous rice husk silica through branched polyethyleneimine grafting for phosphate adsorption. ND Suzaimi, PS Goh, NANN Malek, JW Lim, AF Ismail. Arabian Journal of Chemistry 13 (8), 6682-6695

Experimental and modeling of dicamba adsorption in aqueous medium using MIL-101 (Cr) metal-organic framework. HA Isiyaka, K Jumbri, NS Sambudi, JW Lim, B Saad, A Ramli, ZU Zango.Processes 9 (3), 419

 U have to ensure the unit used is consistent. Are u using / or -1 for the denominator?

The R2 value is best for Pseudo-first order?

The deltaG value seem insignificant among all the studies temperature. Perhaps, this needs explanation please.

Isotherms study needs more point in Fig 5.

Table 5 is simple and can be embedded in the text for the values.

Author Response

Reviewer 2

Comments and Suggestions for Authors

  1. Abstract: Propranol • HCl –This is an error?

Reply:

It is an error. The commercial corrected name of the used pollutant is Propranolol • HCl, which has been corrected throughout the manuscript.

  1. Abstract lacks of numerical values that are important.

Reply:

The manuscript has been revised accordingly.

  1. How PRO is treated now and what is the limitations of the treatment mode? Please elaborate in Introduction.

Reply:

Table 1 already reported a summary of the most important works available in the literature about PRO removal. The whole manuscript has been revised according to this comment to emphasize better the positive aspects proposed with our work with respect to the past literature.

  1. Section 2.2 is simple. Please elaborate how the adsorbent was prepared in detail. Grinding, size, etc.

Reply:

The manuscript has been revised accordingly. In detail, Kiwi Peels were placed in hot water until the fruit pulp was removed and left to dry in an oven at 50°C. Subsequently, the latter were grounded to obtain homogeneous shaped and sized Peels. Certified test sieves (Giuliani, series ASTM, American Society for Testing and Materials International) were used for this purpose. Peels having a size of 5 mm x 5 mm were selected and used during this work

  1. Eq 2 and 3 the K should be k.

Reply:

The manuscript has been revised accordingly.

  1. Please check other Eqs carefully and please refer these papers and cite if needed (you can refer other papers too): Artificial neural network (ANN) for modelling adsorption of lead (Pb (II)) from aqueous solution.T Khan, MRU Mustafa, MH Isa, TSBA Manan, YC Ho, JW Lim, NZ YusofWater, Air, & Soil Pollution 228 (11), 1-15

Performance of branched polyethyleneimine grafted porous rice husk silica in treating nitrate-rich wastewater via adsorption. ND Suzaimi, PS Goh, NANN Malek, JW Lim, AF Ismail. Journal of Environmental Chemical Engineering 7 (4), 103235

Enhancing the performance of porous rice husk silica through branched polyethyleneimine grafting for phosphate adsorption. ND Suzaimi, PS Goh, NANN Malek, JW Lim, AF Ismail. Arabian Journal of Chemistry 13 (8), 6682-6695

Experimental and modeling of dicamba adsorption in aqueous medium using MIL-101 (Cr) metal-organic framework. HA Isiyaka, K Jumbri, NS Sambudi, JW Lim, B Saad, A Ramli, ZU Zango.Processes 9 (3), 419

Reply:

The suggested manuscripts are very interesting, and all of them have been used and cited to improve the quality of the manuscript. The manuscript has been revised accordingly.

  1. U have to ensure the unit used is consistent. Are u using / or -1 for the denominator?

Reply:

We thank the Reviewer for this suggestion. The manuscript has been revised accordingly.

  1. The R2 value is best for Pseudo-first order?

Reply:

Yes, as described in the text, the R2 values and the comparison between qe,exp/qe,calc were investigated to select the best model able to fit the data. See the revised text for more details. 

  1. The deltaG value seem insignificant among all the studies temperature. Perhaps, this needs explanation please.

Reply:

We apologize. In Table 4, there were typos associated with Delta G values. Table 4 has been revised accordingly.

  1. Isotherms study needs more point in Fig 5.

Reply:

The number of experiments has been increased; Figure 5 has been revised accordingly.

  1. Table 5 is simple and can be embedded in the text for the values.

Reply:

Ok, Table 5 has been removed, and the values have been inserted in the main text.

Reviewer 3 Report

The present MS deals with adsorption/desorption processes of emerging pollutants using organic wastes as adsorbent material. The topic seems to be attractive as a wastewater treatment option. Nevertheless, some comments should be addressed before its consideration for publication:

1)    In the introduction section, the weaknesses of the adsorption process should be mentioned as the disadvantages of the other treatment processes. 

2)    Line 11, is it propanol or Propranolol? Please revise all the MS for typos

3)    Revise references in lines 207 and 208.

4)    The schematic part of Figure 1 must be presented as graphical abstract, and only the UV-Vis spectra should be shown in Fig. 1. 

5)    Simplify the figures' captions.

6)    Line 358, which figure do you refer to?

7)    Revise references in lines 381 and 382.

8)    In Table 2, the R2 parameter for 10 and 15 ppm is relatively low. Please explain as in Table 3 with 25 mg for Kiwi peel.

9)    The data in Table 5 could be presented as text.

10) If the authors have some mentions about Circular Economy and Green Chemistry principles, they should be mentioned a little bit more in the introduction section. 

11) How do the Kiwi peels after the treatments are finally disposed?

12) The Kiwi adsorbent will be part of a packed Column? Why did the authors decide to use it in a suspension? The retention time should be calculated.

Author Response

Reviewer 3

Comments and Suggestions for Authors

The present MS deals with adsorption/desorption processes of emerging pollutants using organic wastes as adsorbent material. The topic seems to be attractive as a wastewater treatment option. Nevertheless, some comments should be addressed before its consideration for publication:

  • In the introduction section, the weaknesses of the adsorption process should be mentioned as the disadvantages of the other treatment processes. 

Reply:

The manuscript has been revised accordingly.

  • Line 11, is it propanol or Propranolol? Please revise all the MS for typos

Reply:

The manuscript has been revised accordingly.

  • Revise references in lines 207 and 208.

Reply:

The manuscript has been revised accordingly.

  • The schematic part of Figure 1 must be presented as graphical abstract, and only the UV-Vis spectra should be shown in Fig. 1. 

Reply:

The manuscript has been revised accordingly.

  • Simplify the figures’ captions.

Reply:

The captions have been revised accordingly.

  • Line 358, which figure do you refer to?

Reply:

The manuscript has been revised accordingly. The Figures are 3A and 3B.

  • Revise references in lines 381 and 382.

Reply:

We apologize for this inconvenience. It is an error due to the use of software (Mendeley) to insert references. The manuscript has been revised accordingly.

  • In Table 2, the R2parameter for 10 and 15 ppm is relatively low. Please explain as in Table 3 with 25 mg for Kiwi peel.

Reply:

The manuscript has been revised accordingly.

  • The data in Table 5 could be presented as text.

Reply:

The manuscript has been revised accordingly.

  • If the authors have some mentions about Circular Economy and Green Chemistry principles, they should be mentioned a little bit more in the introduction section. 

Reply:

We agree with the Referee’s suggestion. The manuscript has been revised accordingly to evidence these aspects better.

  • How do the Kiwi peels after the treatments are finally disposed?

Reply:

This work aims to avoid the disposal of the adsorbent. Indeed, the possibility of recycling both the pollutant and the adsorbent employing salt solutions has been demonstrated. The materials could be considered very performant and, as demonstrated by Gubitosa et al.[31], the features of the adsorbent are also retained after several runs of adsorption and desorption, opening the concrete possibility to propose this material for real applications, avoiding its disposal. Currently, 10 cycles of adsorption and desorption have been performed, evidencing that Kiwi Peels retained its performance in terms of % of adsorption; however, the number of cycles can be increased. So, this work intends to propose a circular approach to avoid waste dumping, lowering the environmental impact. Anyway, at the end of process (that at moment cannot be supposed in term of time/use because the material works well although its prolonged use), after the PRO desorption, to avoid problem related to environmental pollution, the material could be treated as a simple waste, following the classical procedures to dispose it. These aspects have been emphasized in the revised paper.

  • The Kiwi adsorbent will be part of a packed Column? Why did the authors decide to use it in a suspension? The retention time should be calculated.

Reply:

Preliminary in-flux experiments have been performed by using a packed column. Indeed, Gubitosa et al. [31] recently demonstrated the possibility of working under dynamic experimental conditions by using Kiwi Peels. A commercial syringe-like column (3.0×5.0 cm) was filled with Kiwi Peels and swelled with water (see Figure 1 reported in the following). A solution of PRO (10 mg/L) was slowly flowed through the column by a piston. The eluted solution was collected and measured by UV-Vis spectroscopy. After 30 minutes of elution, the % of PRO adsorption was calculated, resulting in about 50% removal. It is worth mentioning that important counter pressures were absent during the experiments, even if measurements were performed for a prolonged time. This information has been added to the revised paper as preliminary results because it must be optimized.

So, the Authors have mainly decided to work in batch mode with the aim of inferring kinetic and thermodynamic information related to the adsorption process. In our opinion, the knowledge of the physical and chemical features of the adsorbent and the whole adsorption process is a very important step before performing in-flux experiments. The idea is to show the best conditions of work to be used, for example, for industrial applications. We are working in this direction, intending to build a lab prototype by using packed columns.

Figure 1: Camera picture of the syringe-like column used for performing flow experiments to study the PRO adsorption onto Kiwi Peels under dynamic conditions.

Reviewer 4 Report

The manuscript investigated the adsorption process of Propranol • HCl on Kiwi peels due to the low cost of kiwi peels. The idea is interesting and the study is well organized. But the major problem is the low adsorption capacity. It takes longer time to reach equilibrium compared to the reported adsorbents. The author mentioned the cycling process could compensate the low adsorption capacity, however, the time and cost spending in the cycling process will be much higher than that of Kiwi Peels. It is not economic to treat low-cost Kiwi Peels with such complicated processes to improve the adsorption capacity.  So I reject the publication of this manuscript.

Author Response

Reviewer 4

Comments and Suggestions for Authors

The manuscript investigated the adsorption process of Propranol • HCl on Kiwi peels due to the low cost of kiwi peels. The idea is interesting and the study is well organized. But the major problem is the low adsorption capacity. It takes longer time to reach equilibrium compared to the reported adsorbents. The author mentioned the cycling process could compensate the low adsorption capacity, however, the time and cost spending in the cycling process will be much higher than that of Kiwi Peels. It is not economic to treat low-cost Kiwi Peels with such complicated processes to improve the adsorption capacity.  So I reject the publication of this manuscript.

Reply:

We acknowledge the Referee for the time dedicated to our paper and for His/Her positive comments about the idea and organization of this work. However, we don’t agree with the Referee’s negative comments. We are surprised about His/Her explanation for rejecting the paper. The low adsorption capacity, in our opinion, cannot be the right motivation to reject a paper. Indeed, the very important aspect of this paper is related to the adsorbent and pollutant recycling, valorizing food wastes according to Green Chemistry and Sustainability principles.

Indeed, using Kiwi Peels could offer great advantages through reutilizing waste and obtaining high gains. Kiwi cultivation occupied a surface of ~ 270 × 103 ha around the world, and Europe contributed with ~ 43 × 103 ha. Accordingly, the Kiwi waste production is high, accounting for ~ 4.5 × 106 tons globally, with Europe producing ~1 × 106 tons. Consequently, this implies a huge global production of Kiwi wastes with dangerous consequences for the environment, encouraging the proposed circular economy approach as the most pertinent.

The green and sustainable approach proposed for the adsorbent and pollutant recovery (by using salt solutions) further valorizes the waste obtaining an important gain.

So, although the maximum adsorption capacity is relatively low, it is possible to increase this value by performing several adsorptions and desorption cycles. In this case, we have demonstrated the possibility of performing 10 cycles of adsorption/desorption increasing the maximum adsorption capacity from 2 to 20 mg/g, a value that could be further improved. Indeed, as demonstrated in our previous work (see reference 31), the Peels features appeared unchanged after the recycling, favoring their potential application for industrial purposes (work is in progress in this direction in our laboratory). Moreover, in our recent manuscript (see reference 31), it has been demonstrated the possibility of also removing other different pollutants present in complex mixtures. Furthermore, we are working on the possibility also to adsorb textile dyes and heavy metals for wide-ranging applications.

About the time and cost spending in the cycling process that, according to the Reviewer, are much higher than that of Kiwi Peels, again, we don’t agree with His/Her evaluation. The time necessary for the regeneration is short, and the use of salts for the desorption, in our opinion, is not expensive. The Authors selected MgCl2. However, NaCl could also be used, and seawater could be a valid alternative to reduce costs further. As demonstrated by Gubitosa et al. [31] the adsorbent is long-lasting. Therefore the associated costs should be only ascribed to its preparation. The pollutant is also recovered, obtaining an additional gain in terms of costs.

In particular, to evaluate the associated costs, the following considerations have been taken in consideration: (i) the cost associated with the raw adsorbent could be considered null because it is a waste received from agri-food industries; (ii) only the costs associated with the washing hot water, needed for the pre-treatments of Kiwi Peels, should be considered. To wash 1Kg of Peels, around 300L of water are necessary, and the supposed cost is around 0.42 € (the cost for 1m3 of tap water can supposed be 1.4 €). The important contribution of heat necessary to warm water is further considered (around 0.18€ are needed to heat 1L of water), and 54 € can be necessary to prepare 1Kg of adsorbent. On this ground, the total cost arises, and it can be mainly attributed to the heat necessary for heating water useful for pre-treating the adsorbent before its use. However, the cost is very low compared to the cost of active carbon (around 300 € per Kg) and can be easily sustained by industries, considering that the additional positive aspect is related to the adsorbent recycling with salt solutions. Moreover, a gain could also come from the PRO recovering (the retail cost of PRO is about 30 € per g) that, after appropriate controls, could be reintroduced in the market.

If the process, according to the Referee, is expensive, the gain for the industry is related to the fact that a large number of compounds can be removed from water (see text and reference 31) by using a low-cost material.

Finally, the use of Kiwi Peels will favor a new virtuous alliance between agriculture, enterprise, and research. So, the academic research would close the bioeconomic circle, giving rise to promising collaborations that will multiply the value of Kiwi fruits while improving the engineering of production processes with the development of new plant technologies for the transformation of Kiwi wastes into value-added materials. In turn, this innovation will enhance the competitiveness of Italian producers and revitalize the territory.

These considerations have been better emphasized in the revised manuscript to highlight the aim of this work and why Kiwi Peels can be considered valid and competitive with other materials.

In our opinion, all these positive aspects can overcome the low adsorption capacity, thus evidencing the importance of this work.

Reviewer 5 Report

The manuscript entitled “Regenerable Kiwi Peels as adsorbent to remove and reuse the emerging pollutant Propranolol from water” investigates the use of Kiwi Peels as an adsorbent for propranolol from water. The authors studied the physical and chemical characteristics of the adsorption process, like the effect of ionic strength, pH values, adsorbent/pollutant amounts, and temperature values. The thermodynamics, the adsorption isotherms, and the kinetics of the adsorption process were also carefully investigated. This topic is very important, and the manuscript is prepared very well. Still, my major concern is the lack of novelty. The same group of the author published in 2022. the use of the same material as an adsorbent for ciprofloxacin (ref 31 in the manuscript). This manuscript looks just like ref 31. In my opinion, the results would have to be presented differently to vary from the previously published paper. At least some kind of new insight into the process should be provided. Also, the extensive self-citation of the authors is noticeable. I suggest introducing the investigation of some real samples to give the manuscript some variation with respect to ref 31.

Line 432: Why as expected? It is hardly expected that material originating from the biomass follows the strict demands of the Langmuir model.

Table 5 is unnecessary; the data could be given within the text.

Author Response

Reviewer 5

Comments and Suggestions for Authors

The manuscript entitled “Regenerable Kiwi Peels as adsorbent to remove and reuse the emerging pollutant Propranolol from water” investigates the use of Kiwi Peels as an adsorbent for Propranolol from water. The authors studied the physical and chemical characteristics of the adsorption process, like the effect of ionic strength, pH values, adsorbent/pollutant amounts, and temperature values. The thermodynamics, the adsorption isotherms, and the kinetics of the adsorption process were also carefully investigated. This topic is very important, and the manuscript is prepared very well. Still, my major concern is the lack of novelty. The same group of the author published in 2022. the use of the same material as an adsorbent for ciprofloxacin (ref 31 in the manuscript). This manuscript looks just like ref 31. In my opinion, the results would have to be presented differently to vary from the previously published paper. At least some kind of new insight into the process should be provided. Also, the extensive self-citation of the authors is noticeable. I suggest introducing the investigation of some real samples to give the manuscript some variation with respect to ref 31.

Line 432: Why as expected? It is hardly expected that material originating from the biomass follows the strict demands of the Langmuir model.

Table 5 is unnecessary; the data could be given within the text.

Reply:

We acknowledge the Referee for the time dedicated to our paper, and for His/Her positive comments about the idea and organization of this work. The very important aspect of this paper is related to the adsorbent and pollutant recycling, valorizing food wastes according to Green Chemistry and Sustainability principles. Indeed, using Kiwi Peels could offer great advantages in terms of the eco-sustainability of decontamination processes.

Moreover, notwithstanding our recent article describing the application of Kiwi Peels as a sustainable adsorbent, the number of publications regarding the exploitation of this resource is very low. Even fewer are studies concerning the removal of Propranolol (see Table 1 in the paper). Therefore, in our opinion, although it is not possible to define this paper as very innovative, still, it better characterizes the behavior of this waste/bioresource, enhancing its real potential application and demonstrating its very good performance.

Furthermore, we are working on the possibility also to adsorb textile dyes and heavy metals for wide-ranging applications.

Finally, the use of Kiwi Peels will favor a new virtuous alliance between agriculture, enterprise, and research. So, the academic research will close the bioeconomic circle, giving rise to promising collaborations that will multiply the value of Kiwi fruits, while improving the engineering of production processes with the development of new plant technologies for the transformation of Kiwi wastes into value-added materials. In turn, this innovation will enhance the competitiveness of Italian producers and revitalize the territory.

This information has been better emphasized in the revised manuscript to highlight the aim of this work and why Kiwi Peels can be considered valid and competitive with other materials.

About Line 432, the Langmuir model was expected because in the first part of the results and discussion, the proposed hypothesis and observed results suggested that the Langmuir model was probably the best for describing the adsorption process. Please, see the details in the revised paper. 

Table 5 has been removed.

Round 2

Reviewer 1 Report

The paper can be accepted

Author Response

Reviewer 1

Comments and Suggestions for Authors

The paper can be accepted

Reply:

We thank the Referee for the positive comment and for giving us the possibility of improving the quality of the manuscript for presenting in literature the obtained results.

Reviewer 4 Report

Please pay attention to the format of the figures, using uniform fond and size. Make sure that the positions of number A B C in the figures are clear to the readers. 

Author Response

Reviewer 4

Comments and Suggestions for Authors

Please pay attention to the format of the figures, using uniform fond and size. Make sure that the positions of numbers A B C in the figures are clear to the readers.

Reply:

The Figures have been revised accordingly.

Reviewer 5 Report

Although the authors refused to address my comments, I see their point and recommend the manuscript for publication in its present form. It is important to have more papers on this topic. Still, by not taking the time and effort to implement the alterations, the authors lost the chance to publish an outstanding manuscript. Instead, here is a mediocre text on an already seen topic.  

Author Response

Reviewer 5

Comments and Suggestions for Authors

Although the authors refused to address my comments, I see their point and recommend the manuscript for publication in its present form. It is important to have more papers on this topic. Still, by not taking the time and effort to implement the alterations, the authors lost the chance to publish an outstanding manuscript. Instead, here is a mediocre text on an already seen topic. 

Reply:

We thank the Referee for His/Her comments and for giving us the possibility to present in literature the obtained results.